

# EFR-FCOS: enhancing feature reuse for anchor-free object detector

Yongwei Liao, Zhenjun Li, Wenlong Feng, Yibin Zhang and Bing Zhou

School of Information and Communications Technology, Shenzhen City Polytechnic, Shenzhen, Gangdong, China

## ABSTRACT

In this paper, we propose enhancing feature reuse for fully convolutional one-stage object detection (EFR-FCOS) to aim at backbone, neck and head, which are three main components of object detection. For the backbone, we build a global attention network (GANet) using the block with global attention connections to extract prominent features and acquire global information from feature maps. For the neck, we design an aggregate feature fusion pyramid network (AFF-FPN) to fuse the information of feature maps with different receptive fields, which uses the attention module to extract aggregated features and reduce the decay of information in process of the feature fusion. For the head, we construct a feature reuse head (EnHead) to detect objects, which adopts the cascade detection by the refined bounding box regression to improve the confidence of the classification and regression. The experiments conducted on the COCO dataset show that the proposed approaches are extensive usability and achieve significant performance for object detection.

## INTRODUCTION

According to the framework of current object detection algorithms, which includes two categories: convolutional neural network (CNN) based (*Krizhevsky, Sutskever & Hinton, 2012*; *He et al., 2016*) and Transformer-based (*Vaswani et al., 2017*) detection algorithms. DERT (*Carion et al., 2020*) constructs a fully end-to-end object detection framework based on transformer, but DERT requires the longer time to train and higher resource consumption than CNN-based Faster R-CNN (*Ren et al., 2015*). In particular, ViT-FRCNN (*Beal et al., 2020*) replaces the convolutional neural network with transformer as the backbone for feature extraction. Although the Transformer-based detection algorithms achieve certain advantages in accuracy, they require significant computational resources and time to train. Therefore, it is still too early to completely replace the convolutional neural network with transformer.

One-stage object detection (*Tian et al., 2019*) is a representative CNN-based methodology, which is generally designed to recognize the object classification and location directly through the backbone network. It includes the anchor-based methods such as single-shot detector (SSD) (*Liu et al., 2016*), RetinaNet (*Lin et al., 2017b*), Cascade RetinaNet (*Zhang et al., 2019a*), *etc.*, and the anchor-free methods (*Tian et al., 2020*) such

Corresponding author
Bing Zhou, bingzhou6@gmail.com

as DenseBox (*Huang et al., 2015*), YOLO (*Hussain, 2024*), CornerNet (*Law & Deng, 2020*), fully convolutional one-stage object detection (FCOS) (*Tian et al., 2019*), RepPoints (*Yang et al., 2019*), *etc.* Moreover, the anchor-free methods avoid the complicated computations caused by using extensive anchor boxes and reduce the design of hyper-parameters and have become the popular method.

The structure of one-stage object detector consists of backbone, neck and head. The popular backbone networks include VGG (*Simonyan & Zisserman, 2015*), ResNet (*He et al., 2016*), Transformer (*Vaswani et al., 2017*), *etc.* In particular, ResNet is the most widely used due to its simple structure and less parameters, which introduces an identity skip connection to alleviate vanishing gradient and allows network to learn deep feature representations effectively. From the bottleneck block of ResNet, the information transmitted directly from the previous layer to the next layer usually experiences a certain attenuation. Therefore, we design a global attention network (GANet), which refines the information and gain global features from feature maps.

The neck is used to gather multi-scale features from various stages of the backbone network, such as feature pyramind network (FPN) (*Lin et al., 2017a*), Adaptively Spatial Feature Fusion (ASFF) (*Liu, Huang & Wang, 2019*), Receptive Field Block (RFB) (*Liu, Huang & Wang, 2018*), spatial pyramid pooling (SPP) (*He et al., 2015*), *etc.* However, the current FPN-based methods may drop features when fusing multi-scale features directly. Therefore, we propose an aggregate feature fusion pyramid network (AFF-FPN) for fusing the feature from feature maps with different receptive fields, which can extract global context and reduce the decay of information to learn more comprehensive feature representation.

The head is used to detect the locations and classify the objects, which generally is composed of a detection network, bounding boxes and loss. In particular, the FCOS-based point-wise prediction introduces the branch of center-ness to suppress low-quality predicted bounding boxes for improving the performance of the detector. In addition, the cascade detection is verified by Cascade R-CNN (*Cai & Vasconcelos, 2018*) and Cascade RetinaNet (*Zhang et al., 2019a*), which are an effective mechanism. Inspired by this, we propose the EnHead of cascade detection mechanism to improve the detection performance remarkably.

In summary, the performance of object detection benefits from enhancing feature extraction, improvement of tiny targets, refines the information and gain global features. We use multi-scale feature extraction and feature fusion to improve the performance of the detector, and cascade detection to improve the detection for tiny targets. In this paper, we design a global attention network (GANet) to enhance the capability of feature extraction, which can gain global information to strengthen feature extraction. To reduce the decay of information, we design an aggregate feature fusion pyramid network (AFF-FPN) to fuse multi-scale feature information from feature maps. Motivated by the cascade detection, to achieve the refinement of bounding boxes, we design the feature reuse head (EnHead) for improving the confidence of regression.

The main contributions of our work are summarized as follows:

- We propose a novel global attention network as the backbone of object detector (denoted as GANet), which aims to collect salient information for improving the extraction capability of multi-scale features.
- We propose an aggregate feature fusion pyramid network as the neck of object detector (denoted as AFF-FPN), which aims to fuse multi-scale feature information by extracting global context.
- We propose a novel feature reuse head as the head of object detector (denoted as EnHead) for the anchor-free one-stage method, which aims to improve the confidence of regression by refining the bounding boxes.

The organization of this paper is as follows. We start our discussion with a brief review of improving on the object detection in 'Related Work'. 'Methodologies' presents our approaches. 'Experiments' evaluates the methodologies. 'Conclusion' concludes the paper.

## RELATED WORK

As our work focuses on improving feature learning for object detection, we investigate representative works of improvements on backbone, neck and head of object detector.

### Improvements on ResNet

As a component of the object detectors, the backbone network is used to extract features, and designing a neural network architecture is a fundamental task of computer vision. The representative works of deep neural network includes (*Simonyan & Zisserman, 2015*), ResNet (*He et al., 2016*), Dilated Convolution (*Yu & Koltun, 2015*), EfficientNet (*Tan & Le, 2019*) and ViT (*Dosovitskiy et al., 2021*) based on transformer. In particular, ResNet is the common neural network in the tasks of computer vision, which uses skip connections to alleviate the vanishing-gradient problem when constructing the deep network. Because it has few parameters, simple structure and can be easily used as a backbone for other visual tasks, and has become one of the most popular architecture. At present, the improved versions of ResNet focus on improving the trunk of the residual block and adding the attention module to residual block, such as DenseNet (*Huang et al., 2017*), ResNeXt (*Xie et al., 2017*), Res2Net (*Gao et al., 2021*), SENet (*Hu, Shen & Sun, 2017*), GCNet (*Cao et al., 2019*), TripleNet (*Misra, Nalamada & Landskape, 2020*). DenseNet proposes a dense convolutional networks, where skip connections pass the feature maps of the former layers to each latter layer. adopt group convolution to increase cardinality. Res2Net constructs hierarchical residual-like connections within one single residual block for stronger multi-scale representation ability. SENet adopts channel-wise rescaling to explicitly model dependency among channels, which presents an effective mechanism to learn channel attention and achieves promising performance. GCNet maintain the accuracy of NLNet while significantly reducing computational complexity. TripleNet proposes triplet attention of three-branch structure, which builds inter-dimensional dependencies by the rotation operation followed by residual transformations and encodes inter-channel and spatial information. Unlike these networks focus on the trunk of the residual block, we focus on skip connections and propose a global attention network (GANet), which can be plugin into residual block for extracting salient information from feature maps.

## Improvement on FPN

Generally, the neck of object detector is used to fuse the feature from multi-scale feature maps, the popular network is FPN (*Lin et al., 2017a*), PANet (*Liu et al., 2018*), *etc.* In particular, FPN has been the most effective and important structure to extract multi-scale features for object detector, and all levels of feature maps contain strong semantic information. PANet adds a bottom-up channel on the basic of FPN, which by using the accurate low-level positioning signals to enhance the feature representation for shortening the path of information between the low-level and top-level features. By improving multi-scale features with strong semantics, the performance of object detection has been substantially improved, such as BFPN (*Wu et al., 2018*), Bi-FPN (*Tan, Pang & Le, 2020*), PFPN (*Wang et al., 2020a*), AugFPN (*Guo et al., 2020*), iFPN (*Wang, Zhang & Sun, 2021*), *etc.* Bi-FPN adds a cross-scale connection (residual connection) on the basis of PANet to obtain a more advanced feature fusion. CE-FPN (*Luo et al., 2020*) proposes a sub-pixel skip fusion method to perform both channel enhancement and up-sampling. However, FPN-based methods suffer from the inherent defects of channel reduction, which leads to the loss of semantic information and information decay during fusion. Therefore, we propose enhanced feature pyramid network (AFF-FPN) to fuse the feature of local and global receptive fields for improving feature representation.

## Improvements on Head

In the terms of head, for classification and regression of object detection, which is generally composed of detection network, bounding boxes and loss. Compare with the anchor-based method, the anchor-free method avert the complicated computation of using anchor boxes and reduce the hyper-parameters. In anchor-free keypoint-based approaches of using Hourglass as the backbone, CornerNet (*Law & Deng, 2020*) uses a pair of the top-left corner and the bottom-right corner of keypoints to detect bounding boxes. ExtremeNet (*Zhou, Zhuo & Krahenb, 2019*) uses a standard keypoint estimation network to detect four extreme points of top-most, left-most, bottom-most, right-most, and one center point of objects. CenterNet (*Zhou, Wang & Krahenb, 2019*) uses a triple of keypoints of the top-left corner, the bottom-right corner, and a center point to detect bounding box, which enriching information collected by both top-left and bottom-right corners, and providing more recognizable information at the central regions. Moreover, the anchor-free approach FSAF (*Zhu, He & Savvides, 2019*) utilizes ResNet as backbone, the proposed FSAF module involves the online feature selection applied to the training of multi-level anchor-free branches, which is attached to each level of the feature pyramid. Free-Anchor (*Zhang et al., 2019b*) modifies the loss function to avoid manually assign anchors, which can learn anchors matching with the objects. FoveaBox (*Kong et al., 2020*) based on ReintaNet, multi-scale objects are assigned to different feature layers for classification and regression on pixels directly. FCOS (*Tian et al., 2019*) introduces the branch of center-ness to suppress low-quality predicted bounding boxes for improving the performance of the detector. FCOS v2 (*Tian et al., 2020*) moves the branch of Center-ness from classification to regression for prediction. In particular, the improvement of FCOS such as NAS-FCOS (*Wang et al., 2020b*), TPS-FCOS (*Sun et al., 2020*) *etc.*, they all use single detection head. In addition,

the detection network of the head can be divided into single detection for classification and bounding box regression such as Faster R-CNN (*Ren et al., 2015*), SSD(*Liu et al., 2016*), RetinaNet (*Lin et al., 2017b*), FCOS (*Tian et al., 2019*), *etc.*, and cascaded detection utilizes feature sharing to ensemble multi-stage outputs such as Cascade R-CNN (*Cai & Vasconcelos, 2018*), ConRetinaNet (*Kong et al., 2019*), Cascade RetinaNet (*Zhang et al., 2019a*), *etc.* Motivated by the cascade detection, we propose a feature reuse head (EnHead) to improve the confidence of regression for the refined bounding boxes.

## METHODOLOGIES

We introduce the architecture of EFR-FCOS in this section. The GANet is presented subsequently, followed by AFF-FPN and EnHead.

### EFR-FCOS

In one-stage methods of object detection, anchor-based methods generate dense anchor boxes to increase the recall of objects, which bring out redundant boxes and require to set extensive hyper-parameters such as the scale of the box, aspect ratio of the box, and IOU threshold. In comparison to the anchor-based method, the anchor-free method circumvents the complicated computation caused by using extensive anchor boxes and reduces the design of hyper-parameters, which further towards real-time and high precision of object detection. In addition, keypoints-based methods need to predict multiple keypoints will lead to complex computations such as CornerNet (*Law & Deng, 2020*), CenterNet (*Zhou, Wang & Krahenb, 2019*), *etc.*, while the pixel-wise prediction methods take advantages of all points in a ground truth bounding box to predict the bounding boxes and can provide comparable recall with anchor-based detectors such as FCOS (*Tian et al., 2019*), RepPoints (*Yang et al., 2019*), *etc.* Whether the keypoints-based method or the pixel-wise prediction method, is essentially dense prediction. The vast solution space leads to excessive false positive, which obtains the results of high recall and low precision. For the improvement of anchor-free methods, on the one hand, further improving head with re-weight detection output through various ways. On the other hand, using FPN (*Lin et al., 2017a*) to alleviate the impact of high coincidence. Therefore, we propose an enhancing feature reuse for anchor-free one-stage object detection (name as EFR-FCOS), which makes improvements from the three components of the models for object detection.

Intuitively, as shown in Fig. 1, based on FCOS v2 (*Tian et al., 2020*) due to its simple and effective structure, we propose an enhancing feature reuse for anchor-free object detection, named as EFR-FCOS, consists of three components, *i.e.,* GANet (global attention network), AFF-FPN (aggregate feature fusion for feature pyramid network), and EnHead(enhancing feature reuse for detection head). In terms of the components of the object detector, we propose GANet as the backbone for extracting features, which is similar to the structure of ResNet. It will select layers of conv2, conv3, conv4, and conv5, denoted as {C2, C3, C4, C5}, the strides is {4, 8, 16, 32}, and feed into the subsequent neck network for feature fusion. AFF-FPN as the neck for fusing feature from multi-layer feature maps, which will fuse {C2, C3, C4, C5} and output them as {P2, P3, P4, P5}. EnHead is used to detect the objectds from fused features, which adopts two groups of subnetworks consisting of four

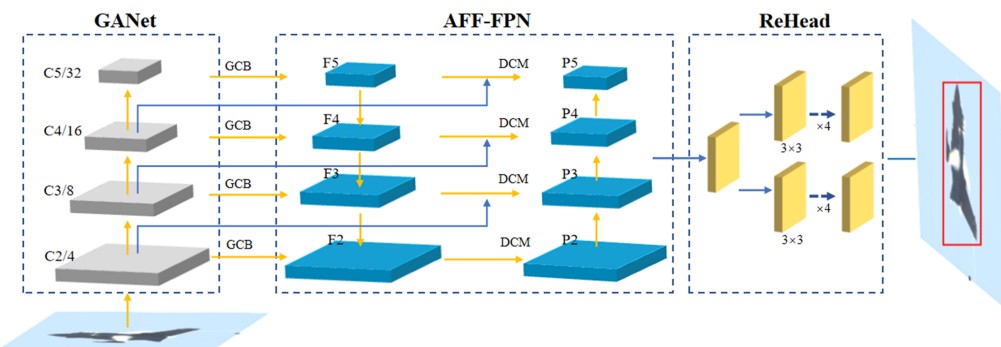

**Figure 1  The structure of EFR-FCOS.**

3×3 convolutions for cascade detection. In addition, similar to FCOS v2, the loss functions adopt FocalLoss, GIoULoss and CrossEntropy for classification, regression and center-ness avoiding a lot of false detection, respectively. The detail will be presented in the following section.

## GANet: global attention network

At present, the prevalent backbone network includes ResNet and Transformer. The Transformer-based network requires huge computational resources and offers slight advantage over ResNet when used for feature extraction. On the contrary, ResNet (*He et al., 2016*) has a simple structure, less parameters, and widely application because it consists of multiple residual modules, *i.e.,* the bottleneck module. In addition, we find that the information transmitted directly from the previous layer to the next layer usually experiences a certain attenuation. Therefore, we develop a global attention network(GANet) module, which can refine the transmitted feature information to capture global features and replace the bottleneck in ResNet.

Intuitively, as shown in Fig. 2, we design the structure of GANet module, which is similar to the residual module of ResNet and consists of a trunk and attention connection. The trunk consists of two 1×1 convolutions, a 3×3 convolution group, and the global context module (GCB) (*Cao et al., 2019*). The attention connection concatenates the average pooling and the max pooling of features from the former layer, followed by a 1×1 convolution, ReLU operation, and Softmax operation, finally fuse with the feature from the trunk. When constructing the backbone network, the method is the same as ResNet. Specifically, we use a set of small 3×3 filters to substitute the 3×3 convolution of the bottleneck in ResNet. After a 1×1 convolution, feature maps are split into groups of subsets, denoted as xi, each subset has the same size, *i.e.,* 1/n. In addition, $F_i(\cdot)$ denotes small 3×3 filters. Therefore, the 3×3 convolution group can be denoted as:

$$F_{i+1} = F_i(x_i + F_{i-1}) \tag{1}$$

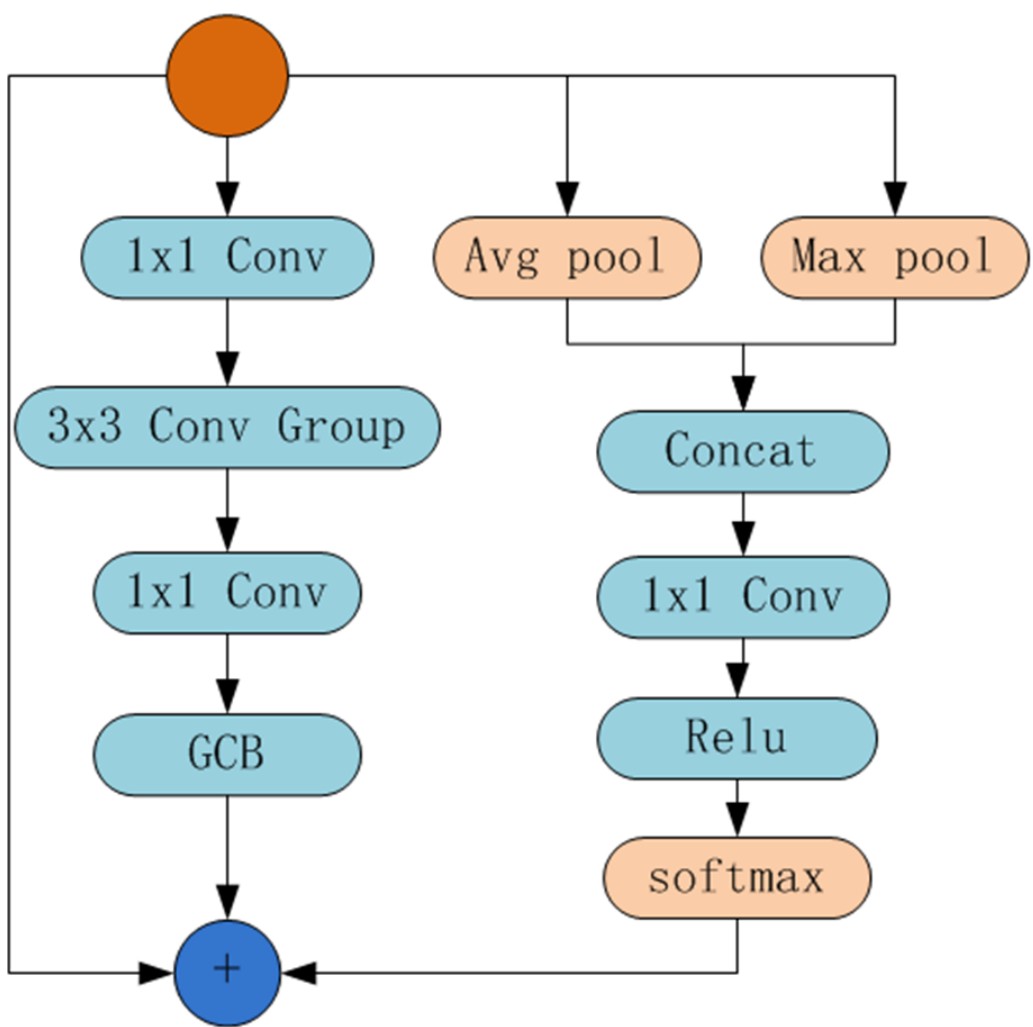

**Figure 2** **The structure of GANet.**

where $x_i$ denotes the feature map. $X$ denotes the imput image, which will through the trunk of GANet module:

$$F(X) = \sum_{1}^{n} F_i(f(x_i)/n) + F_g(x_i) \tag{2}$$

where the $f(.)$ denotes $1 \times 1$ convolution, $F_g$ denotes the global context block, $F$ denotes the output features of the trunk.

In addtion, the another connection, *i.e.*, global attention connection, is used to refine the transmitted feature information except skip-connection of ResNet bypassing the non-linear transformations. Without loss of generality, the output of the global attention connection $G(X)$ is as follows:

$$G(X) = \sum_{1}^{n} S(f(max(x_i) + avg(x_i))) \tag{3}$$

where $avg(.)$ denotes the average pooling, $max(.)$ denotes max function, Relu denotes the rectified linear units, and $S(.)$ denotes Softmax.

In summary, the output of GANet module by integrating the output of the trunk, skip connection, and the global attention connection, which is defined as follows:

$$y = F(X) + X + G(X) \tag{4}$$

where $y$ denotes the output feature maps.

The GANet module in Fig. 2 is the basic unit that makes up GANet, which is similar to the bottleneck in ResNet. Therefore, the constructed backbone network is similar to ResNet. For example, the network of 50 layers includes conv2 consisting of 3 modules, conv3 consisting of 4 modules, conv4 consisting of 6 modules, and conv5 consisting of 3 modules. The network of 101 layers is the same. In particular, conv2, conv3, conv4, and conv5 denoted as C2, C3, C4, C5, the strides is 4, 8, 16, 32, will feed into the subsequent neck network for feature fusion.

## AFF-FPN: aggregate feature fusion for FPN

In terms of the neck, FPN (*Lin et al., 2017a*) can fuse multi-scale features to achieve strong semantic information. By improving multi-scale features with strong semantics, the performance of object detection has been improved significantly, such as Bi-FPN (*Tan, Pang & Le, 2020*), AugFPN (*Guo et al., 2020*), iFPN (*Wang, Zhang & Sun, 2021*), *etc.* However, FPN-based methods suffer from the inherent defect of channel reduction, which leads to the loss of semantic information and information decay during fusion. Therefore, we propose an enhanced feature pyramid network (AFF-FPN) to fuse the feature information of local and global receptive fields for improving feature representation, as shown in Fig. 3. The number of feature maps output by the neck network determines the number of detection heads, but this may not be applicable to various object detectors. FCOS uses various sizes of detection heads to detect objects of different scales and burden is dispersed across multiple feature maps at different levels. In object detectors, FPN will fuse the feature maps of different scales, which generated at different stages from the backbone network (denoted as $C$), and the output features denote as $P$. Therefore, the input of FPN can be denoted as C2 ~C5 and output as P2 ~P5 in the process of feature fusion. The formula is expressed as follows:

$$P_i = H(C_i + C_{i+1}) \tag{5}$$

where $H$ is the feature fusion process.

In AFF-FPN, we introduce three dilated convolutions before feature fusion to reduce the loss of semantic information, and attention module with global context block (GCB) to enhance feature representation for relieving the information decay during fusion. Intuitively, as shown in Fig. 3, following the setting of AFF-FPN, which generates a 4-level feature pyramid. The C2 ~C5 of the backbone output will go through the process of feature fusion. The input image's stride is set to {8, 16, 32, 64}, and the number of channels is {256, 512, 1024, 2048}, respectively. According to the same settings as FPN, {F2, F3, F4, F5} are feature maps with the same number of channels after the global context block (GCB).
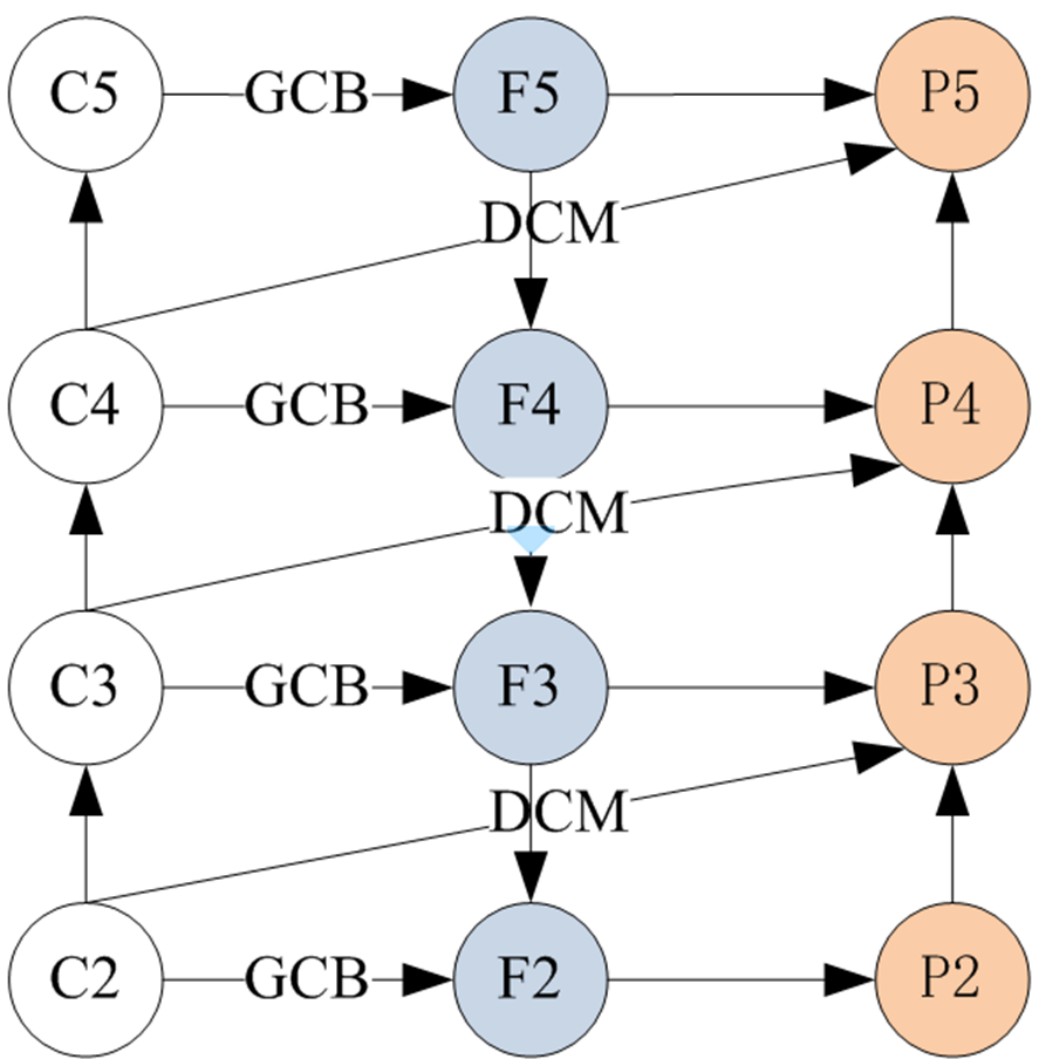

**Figure 3** **The structure of AFF-FPN.**

The feature pyramids {P2, P3, P4, P5} are generated using a top-down path with dilated convolution modules. P6 is obtained directly through the GCB module operation by C5. P7 is obtained through ReLU and a 1×1 convolution on the basis of P6.

Formally, given a list of multi-scale features, AFF-FPN sampling at four different scales of the extracted features from the backbone network. AFF-FPN generates a 4-level feature pyramid {P2, P3, P4, P5} from the output of the backbone {C2, C3, C4, C5}. In top-down feature fusion, the top-down integrates up-sampling features from the latter layer, while GCB extracts global feature information from the $C$ of the backbone. In the bottom-up process, which combines the 3×3 convolution sampling information of this layer, the down-sampling of the previous layer and DCM extracting features at different receptive fields from $C$. The formula of AFF-FPN is defined as follows:

$$F_i^{out} = G(C_i^{in}) + ups(F_{i+1}^{in})$$

$$p_i^{out} = F_i^{in} + D(C_i^{in}) + downs(p_{i-1}^{in}) \tag{6}$$

where $F_i^{out}$ denotes the output of the $i$th layer, $C_i^{in}$ denotes the input of $i$th layer, and $p_i^{out}$ denotes the final output. The *ups* denotes the up-sampling operation, the *downs* denotes down-sampling operation.

Therefore, the structure of AFF-FPN in Fig. 3 is similar to FPN, which selects C2 C5 from the bakcbone network and outputs P2 P5 through feature fusion. It will be feed into the subsequent head network for detecting objects.

## EnHead: Enhancing feature reuse for detection head

The cascade detection is firstly proposed by Cascade R-CNN (*Cai & Vasconcelos, 2018*), which uses cascade regression as a resampling mechanism to increase IoU value stage by stage, so that the resampled proposals from the previous stage can adapt to the next stage with a higher threshold. Cascade RPN (*Vu et al., 2019*) improves the quality of region-proposals and detection performance by systematically addressing the limitation of the conventional RPN that heuristically defines anchors and aligns the feature to the anchor. For one-stage object detection, ConRetinaNet (*Kong et al., 2019*) adopts consistent optimization to match the training hypotheses and the inference quality by utilizing the refined anchor during training. Cascade RetinaNet (*Zhang et al., 2019a*) adopts the idea of cascade for reducing the misalignments of classification and localization. From above mentioned, we find that cascade detection can improve the performance of object detector, which can be applied to EFR-FCOS. Therefore, we design a cascade detecting head for the anchor-free object detection, named as EnHead, which produces high-quality bounding boxes.

RetinaNet (*Lin et al., 2017b*) attaches a small fully convolutional network that consists of four convolutional layers for feature extraction and a single convolutional layer for prediction in different branches. The prediction of each position on the feature map contains classification and regression for various anchor shapes. FCOS (*Tian et al., 2019*) also uses two subnets, which use point-wise to predict boxes and introduce a branch of center-ness to suppress low-quality predicted bounding boxes for improving the performance of detectors. FCOS v2 (*Tian et al., 2020*) moves the branch of center-ness from the branch of classification to the branch of regression for prediction (the following EnHead refers to FCOS v2). In summary, in the head of FCOS, one subnet is used to output the confidence of predicted classification, another for predicting the distance between the center point and the four sides, and the branch of the center-ness for filtering out the point of the low-quality detections. To effectively learn qualified and distributed bounding boxes, we design the EnHead for EFR-FCOS as shown in Fig. 4, which also is used to the improved model based on FCOS.

Intuitively, as shown in Fig. 4, the EnHead based on the cascade mechanism, we refine the bounding boxes, which can obtain high-qualified boxes for improving the performance of detector. The same as FCOS, EnHead includes two subnets, a classification subnet and a box regression subnet for regression and to suppress low-quality boxes. The classification subnet uses only 3×3 convolutions to output classifications and features for boxes, and does not share parameters with the box regression subnet. In parallel with the classification

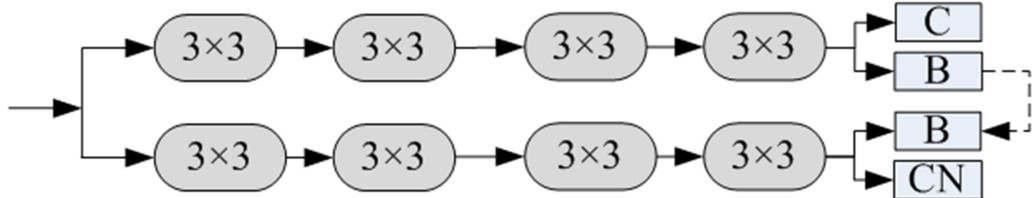

**Figure 4** **The structure of EnHead.**

subnet, we attach four 3×3 convolutions to each pyramid level for regressing the offset from each point to a nearby ground-truth object. In particular, the design of the box regression subnet is different from the classification subnet, which will use the features of location from classification subnet. Since the output is the same as FCOS, the losses will not be changed.

From the mentioned above, we predict the classification scores and regression offsets based on the backbone feature $x_i$. Formally, at the output feature $x_i$ from AFF-FPN, the output feature is the shape of H/4 ×W/4 ×C, where H and W are the input image's height and width. The head performs classification and localization on each grid point of the feature map H/4×W/4 by two parallel convolutional layers. The classification layer predicts the probability of object presence at each grid point for K object categories. The location layer predicts the offset from each grid point to four boundaries of the ground-truth box. The classification subnet can be expressed as:

$$C_i = S(F_i(x_{i,c})) \tag{7}$$

where $x_{i,c}$ indicates the classification of $i$th layer, $C$ indicates the output classification, $S$ indicates Softmax, $F$ indicates convolutions, and the box regression subnet ca be expressed as:

$$R_i = T_k(L(x_{i,r}), L(x_{i,c})) \tag{8}$$

where $x_{i,r}$ indicates the box regression of i-th layer, $L$ indicates the outputting location of boxes, $T_k$ indicates the filtering out $k$ boxes, $R$ indicates the final outputting boxes. It should be noted that the refined offsets are to improve the location representation. From the experiments, we prove that our proposed EnHead can steadily improve the detection performance in different settings.

## EXPERIMENTS

In this section, we will evaluate the proposed GANet, AFF-FPN and EnHead for object detection. The experimental results show that the proposed methods have better performance.

### Dataset and evaluation metrics
We will evaluate the proposed methodologies on the popular MS COCO 2017 (*Tian et al., 2020*) benchmark, which has 80 objects categories and contains about 118k images in the

training set, 5k images in the validation set (*i.e.,* minival), 20k images in test set without the published labels. The average number of objects in a single image of COCO is about three times than VOC (*Ren et al., 2015*), and small objects is majority. In particular, the small objects(less than 32×32) account for 41% approximately, the large objects(greater than 96×96 account for 24% approximately, and the medium objects(range from 32×32 to 96×96 account for 34% approximately. We will train models on the training set and evaluate them on the validation set. In terms of threshold setting, COCO divides it into 10 values, starting with 0.5 and ending with 0.95, with an interval of 0.05. The standard metrics of COCO include $AP, AP_{50}, AP_{75}, AP_S, AP_M, AP_L$, The AP defaults to mAP.

## Implementation details

**Training details**. All models are implemented on the mmDetection wthin PyTorch 1.3 and above. All experiments perform training with an SGD optimizer on four GPUs and two images per GPU, using an initial learning rate of 0.01, a weight decay of 0.0001 and the momentum of 0.9, and using the simplest data augmentation techniques of random cropping, flipping and d stretching. In ablations, most experiments follow the 1×settings where 12 epochs with single-scale training of [800,1333] are used, the learning rate decayed by 10×after epoch 8 and 11. Most of the ablations use ResNet-50 or ResNet-101 as the backbone pretrained on ImageNet. We also use the multi-scale ([480,960]) and longer training (2×settings with 24 epochs in total and the learning rate decay at the epoch of 16 and 22) to test the proposed approaches on stronger backbones, to observe whether the benefits can be maintained on these stronger baselines. In addition, the training schedule is 36 epochs, where the initial learning rate is set to $2.5 \times 10^{-5}$, divided by 10 at epoch 27 and 33, respectively.

**Inference details**. The inference process is quite similar to FCOS (*Tian et al., 2019*). Given an input image, EFR-FCOS directly predicts 100 bounding boxes associated with their scores. The scores indicate the probability of boxes containing an object. For evaluation, we directly use these 100 boxes without any post-processing. Unless specified, we conduct multi-scale testing on the backbone for comparison with the state-of-the-art approaches.Following (*Tian et al., 2019*; *Tian et al., 2020*), we choose the location with $P_{x,y} > 0.05$ as positive samples to obtain the predicted bounding boxes. IoU sets 0.5, 0.6, and 0.7 to select boxes.

## Comparison with the state-of-the-art Methods

Table 1 summarizes the performance of our proposed EFR-FCOS (refer to 'EFR-FCOS') compares with the-state-of-art approaches for object detection. In particular, we select the-state-of-art models of YOLO to compare, include YOLOv8 (*Zhang, Li & Gao, 2024*), YOLOv9 (*Wang, Yeh & Liao, 2024*) and YOLOv10 (*Wang et al., 2024*). Compared to the three versions of M, L and X, our model training is more convenient than the model after YOLOv5, because YOLOv5 uses hyperparameter evolution, freeze training, and image weighting strategy to train. From Table 1 we have some observations: (1) EFR-FCOS achieves the optimal performance, outperforms the recent anchor-free one-stage detector TSP-FCOS (*Sun et al., 2020*) by 3.7%, YOLOv9 and YOLOv10 by 1.3% respectively,

**Table 1  EFR-FCOS vs. the state-of-the-art two-stage and one-stage detectors.**

| Model | Backbone | AP | $AP_{50}$ | $AP_{75}$ | $AP_S$ | $AP_M$ | $AP_L$ |
|---|---|---|---|---|---|---|---|
| Faster R-CNN | ResNet101 | 39.4 | 60.2 | 43.0 | 22.3 | 43.3 | 49.9 |
| CBNet | ResNet-101 | 41.0 | 62.4 | 44.5 | – | – | – |
| TridentNet | ResNet-101 | 42.7 | 63.6 | 46.5 | 23.9 | 46.6 | 56.6 |
| Cascade R-CNN | ResNet-101 | 42.8 | 62.1 | 46.3 | 23.7 | 45.5 | 55.2 |
| SNIP | ResNet-101 | 44.4 | 66.2 | 49.9 | 27.3 | 47.4 | 56.9 |
| Sparse R-CNN | ResNet-101 | 45.6 | 64.6 | 49.5 | 28.3 | 48.3 | 61.6 |
| DDQ | ResNet-101 | 47.8 | 66.3 | 52.6 | 29.9 | 50.0 | 59.3 |
| YOLOv3 | DarkNet-53 | 33.0 | 57.9 | 34.4 | 18.3 | 35.4 | 41.9 |
| YOLOv4 | CSPDarknet53 | 43.0 | 64.9 | 46.5 | 24.3 | 46.1 | 55.2 |
| YOLOv5-S | CSPDarknet53 | 37.4 | 56.8 | – | – | – | – |
| YOLOv6-S | CSPDarknet53 | 44.3 | 61.2 | – | – | – | – |
| YOLOv7-S | CSPVoVNet-53 | 45.1 | 61.8 | 48.9 | 25.7 | 50.2 | 61.2 |
| YOLOv8-S | CSPDarkNet-53(C2f) | 44.9 | 61.8 | – | – | – | – |
| YOLOv9-S | CSPVoVNet-53 | 46.8 | 63.4 | 50.7 | 26.6 | **56.0** | **64.5** |
| YOLOv10-S | CSPDarkNet-53(C2f) | 46.8 | – | – | – | – | – |
| SSD512 | ResNet-101 | 31.2 | 50.4 | 33.3 | 10.2 | 34.5 | 49.8 |
| RetinaNet | ResNet-101 | 39.1 | 59.1 | 42.3 | 21.8 | 42.7 | 50.2 |
| RefineDet | ResNet-101 | 41.8 | 62.9 | 45.7 | 25.6 | 45.1 | 54.1 |
| ATSS | ResNet-101 | 43.6 | 62.1 | 47.4 | 26.1 | 47.0 | 53.6 |
| M2Det | ResNet-101 | 43.9 | 64.4 | 48.0 | 29.6 | 49.6 | 54.3 |
| FreeAnchor | ResNet-101 | 40.9 | 59.9 | 43.8 | 21.7 | 43.8 | 53.0 |
| RepPoints v2 | ResNet-101 | 43.4 | 63.3 | 59.4 | 23.0 | 40.4 | 18.0 |
| FCOS v2 | ResNet-101 | 43.2 | 65.9 | 50.8 | 28.6 | 49.1 | 58.6 |
| TSP-FCOS | ResNet-101 | 44.4 | 63.8 | 48.2 | 27.7 | 48.6 | 57.3 |
| EFR-FCOS | GANet-50 | 45.7 | 66.4 | 52.3 | 29.9 | 50.3 | 58.8 |
| EFR-FCOS | GANet-101 | **48.1** | **66.8** | **53.1** | **30.4** | 50.6 | 60.5 |

**Notes.**
Bold denotes the maximum of each metric in the experiment.

and is comparable to the optimal two-stage detector DDQ (*Zhang et al., 2020*), slightly exceeding 0.3%. This is attributed to the ability of EFR-FCOS, which constructs a new backbone GANet to extract salient and global features from feature maps, while also propose AFF-FPN for fusing global context, and finally design the EnHead to predict the bounding boxes. (2) In addition, for the detection of objects with different sizes, EFR-FCOS can achieve the optimal performance, which indicates the effectiveness of our proposed framework. In particular, extracting the global feature from feature maps and fusing the global context can improve the accuracy of classification and confidence of regression. (3) In the case of the backbone network with same layers, the performance of EFR-FCOS can outperform the current popular detectors, which indicates the effectiveness of AFF-FPN and EnHead and will be explained in the following sections.

## Evaluate the effectiveness of GANet

In Table 2, the memory reported on Titan X GPU. Our proposed GANet outperforms the backbone of ResNet, SENet and ResNeXt, which are trained with the neck of FPN. Based on

**Table 2   Detailed comparison with popular baseline object detectors on COCO 2017.**

| Model | Backbone | GANet | Mem (GB) | AP | $AP_{50}$ | $AP_{75}$ | $AP_S$ | $AP_M$ | $AP_L$ |
|---|---|---|---|---|---|---|---|---|---|
| RetinaNet | ResNet-50 | | 3.8 | 37.4 | 56.3 | 40.3 | 22.2 | 40.9 | 49.7 |
| | ResNet-50 | ✓ | 3.9 | 38.0 | 57.1 | 41.5 | 23.1 | 41.6 | 50.0 |
| | GC-ResNet-50 | | 3.9 | 38.9 | 57.9 | 41.9 | 23.4 | 42.1 | 50.1 |
| | GC-ResNet-50 | ✓ | 4.0 | 39.0 | 58.1 | 42.1 | 22.4 | 42.5 | 50.1 |
| | ResNet-101 | | 5.7 | 39.1 | 59.1 | 42.3 | 21.8 | 42.7 | 50.2 |
| | ResNet-101 | ✓ | 5.8 | 39.7 | 59.5 | 42.6 | 22.7 | 43.1 | 50.4 |
| | GC-ResNet-101 | | 5.9 | 40.2 | 59.9 | 43.5 | 22.9 | 43.3 | 50.9 |
| | GC-ResNet-101 | ✓ | 6.0 | 40.5 | 60.2 | 44.2 | 23.2 | 43.5 | 51.2 |
| | ResNeXt-101-64 ×4d | | 10.0 | 41.0 | 60.7 | 44.7 | 23.4 | 43.5 | 51.9 |
| | ResNeXt-101-64 ×4d | ✓ | 10.1 | 41.3 | 60.9 | 44.8 | 23.7 | 43.8 | 52.1 |
| FCOS | ResNet-50 | | 2.6 | 38.9 | 57.5 | 42.2 | 23.1 | 42.7 | 50.2 |
| | ResNet-50 | ✓ | 2.7 | 39.4 | 58.4 | 43.4 | 24.5 | 44.8 | 51.6 |
| | ResNet-101 | | 5.5 | 43.2 | 62.4 | 46.8 | 26.1 | 46.2 | 52.8 |
| | ResNet-101 | ✓ | 5.6 | 43.8 | 63.2 | 47.3 | 26.8 | 46.8 | 53.4 |
| | ResNeXt-101-64Í4d | | 10.0 | 44.8 | 64.4 | 48.5 | 27.7 | 47.4 | 55.0 |
| | ResNeXt-101-64Í4d | ✓ | **10.1** | **45.2** | **64.6** | **48.7** | **27.9** | **47.8** | **55.2** |

**Notes.**
Bold denotes the maximum of each metric in the experiment.

RetinaeNet and FCOS, we compare the different layers of backbone network to verify the effectiveness of GANet. Table 2 and Fig. 5 summarize the performance of the approaches for object detection, from which we have some observations: (1) The object detectors using the GANet as the backbone network outperform ResNet, GCNet, and ResNeXt. This is reason that GANet can collect salient information for improving the extracting ability of multi-scale features, and verify that the global attention network can be applied to various backbone networks. (2) GANet improves the mAP by about 0.6% to 1.3% on the ResNet. In addition, the detection performance of GC-ResNet with GANet is superior to GC-ResNet, which verifies that GANet can be combined with the block adding the trunk of ResNet. (3) The GANet can improve the performance for large objects of object detectors obviously, because the global attention network can extract global feature information from feature maps.

## Evaluate the effectiveness of AFF-FPN

Based on RetinaNet and FCOS, we compared different FPN-based methods to verify the effectiveness of our proposed AFF-FPN in the Table 3. Among them, BiFPN represents BiFPN-B0. All models were trained with ResNet-50 as the backbone network. Table 3 summarizes the performance of these methods, from which we can observe that: (1) The different models such as RetinaNet and FCOS, which use AFF-FPN as neck, and outperform the existing FPN-based methods, which shows AFF-FPN has generalization and can fuse global semantic information extracting from the backbone network. (2) AFF-FPN is usually superior to the existing FPN-based methods because AFF-FPN can fuse features with spatial and channel information from different receptive fields to improve feature representation ability. Especially in skip connections, a DCM module has been added, which utilizes dilated

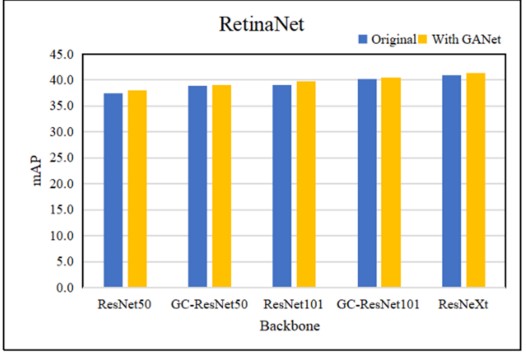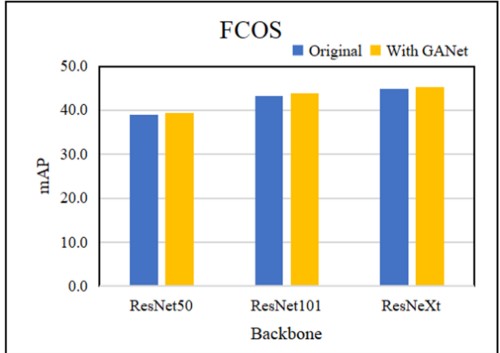

**Figure 5** The result of conduct RetinaNet and FCOS with different backbones.

convolutions of different receptive fields to capture richer global feature information. And during the upsampling process, the fusing network replaces $1 \times 1$ convolutions with the GCB module, which aggregates the features of all positions together to form a global contextual feature for effective modeling of long-distance dependencies. (3) The performance of AFF-FPN has significantly improved for detecting small objects. Because we introduce the global context block(GCB) in AFF-FPN, which can enhance feature representation and reduce information attenuation during feature fusion. Moreover, we design the dilated convolution module with different receptive fields, which can extract abundant global feature information from feature map. (4) For the detection of large objects, AFF-FPN performs the existing FPN-based methods. Because we introduce the dilated convolution module (DCM), which uses dilated convolution kernels of different receptive field, i.e, a sampling rate of {1,3,5}. When we apply $3 \times 3$ convolution kernels with different sampling rates to feature maps, the sampling rate is 1, which is actually a normal $3 \times 3$ convolution. When the sampling rates are 3 and 5, using convolution kernels of the larger sampling rate and can capture global contextual information, which is very effective for extracting feature information from images.

## Evaluate the effectiveness of EnHead

Table 4 summarizes the performance of various object detectors, which is used to verify the effectiveness of EnHead. All models are trained with the backbone of ResNet-101 and FPN. In particular, FCOS-Enhead replace the head network with EnHead, from Table 4 we have some observations: (1) 1n the case of the same backbone and neck, our proposed EnHead is superior to the single head and Double-head of the current one-stage and two-stage object detectors. The reason is that EnHead can improve confidence of classification and regression for exploiting well the feature representation. (2) When the backbone is Resnet-101 and FPN, EnHead exceeds all the latest available methods. AP increased by 1.4%, which indicates that cascade detection is more effective. (3) In addition, EnHead is superior to object detectors with single detection and cascade detection. The reason is that EnHead can improve the ability of detection by selecting the high-score classifications and the top proposals from the output of cascade detection.

**Table 3    Detailed comparison with the popular baseline FPN-based object detectors.**

| Model | Neck | AP | $AP_{50}$ | $AP_{75}$ | $AP_S$ | $AP_M$ | $AP_L$ |
|---|---|---|---|---|---|---|---|
| RetinaNet | FPN | 36.3 | 55.5 | 38.7 | 20.5 | 40.1 | 47.5 |
| | BiFPN | 37.4 | 58.2 | 40.2 | 21.2 | 40.3 | 47.6 |
| | AugFPN | 37.5 | 58.4 | 40.1 | 21.3 | 40.5 | 47.3 |
| | CE-FPN | 37.8 | 57.4 | 40.1 | 21.3 | 40.8 | 46.8 |
| | AFF-FPN | **38.6** | **60.6** | **41.2** | **22.8** | **42.1** | **48.5** |
| FCOS | FPN | 37.0 | 56.6 | 39.4 | 20.8 | 39.8 | 46.4 |
| | BiFPN | 37.4 | 57.4 | 40.1 | 21.1 | 40.0 | 47.1 |
| | AugFPN | 37.9 | 58.4 | 40.4 | 21.2 | 40.5 | 47.9 |
| | CE-FPN | 38.2 | 59.1 | 41.4 | 22.3 | 41.6 | 47.6 |
| | AFF-FPN | **39.2** | **60.7** | **42.6** | **23.9** | **42.7** | **49.2** |

**Notes.**
Bold denotes the maximum of each metric in the experiment.

**Table 4    Comparisons of object detectors results for different algorithms.**

| Model | AP | $AP_{50}$ | $AP_{75}$ | $AP_S$ | $AP_M$ | $AP_L$ |
|---|---|---|---|---|---|---|
| RetinaNet | 39.1 | 59.1 | 42.3 | 21.8 | 42.7 | 50.2 |
| ConRetinaNet | 40.1 | 59.6 | 43.5 | 23.4 | 44.2 | 53.3 |
| Cascade RetinaNet | 41.1 | 60.7 | 45.0 | 23.7 | 44.4 | 52.9 |
| FCOS v2 | 43.2 | 62.4 | 46.8 | 26.1 | 46.2 | 52.8 |
| FCOS-EnHead | **45.1** | **63.8** | **47.6** | **27.9** | **47.8** | **59.7** |

**Notes.**
Bold denotes the maximum of each metric in the experiment.

The ROC curve is suitable for evaluating the overall performance of classifiers due to its balance between positive and negative examples, while the PR curve completely focuses on positive example. To validate the loss function and evaluate the recall of object detection methods, we compare the recall curves of RetinaNet, FCOS, and EFR-FCOS. Figure 6 shows the recall curves at IOU thresholds of 0.50, 0.75, and 0.90, respectively. From which we can observe that: (1) EFR-FCOS can achieve better performance than the anchor-based counterpart RetinaNet and the anchor-free counterpart FCOS. The reason is that EnHead can achieve high-quality bounding boxes by refining the bounding boxes. (2) It is worth noting that through stricter IoU thresholds, EFR-FCOS has a greater improvement than RetinaNet and FCOS, indicating that EFR-FCOS has better bounding box regression and can detect objects more accurately. The reason is that EFR-FCOS has the ability to use the massive positive examples (3) As shown in all precision–recall curves, the best recalls of these detectors in the precision–recall curves are between 95% and 99.5%.

## Ablation experiments

Table 5 compares the detection performance on COCO datasets with different IoU thresholds set in the second stage. From the Table 5, it can be seen that simply adding a new stage with the same IoU setting will not improve detection accuracy. For EFR-FCOS with an IoU threshold of 0.5, the AP remains unchanged. We believe that the main reason

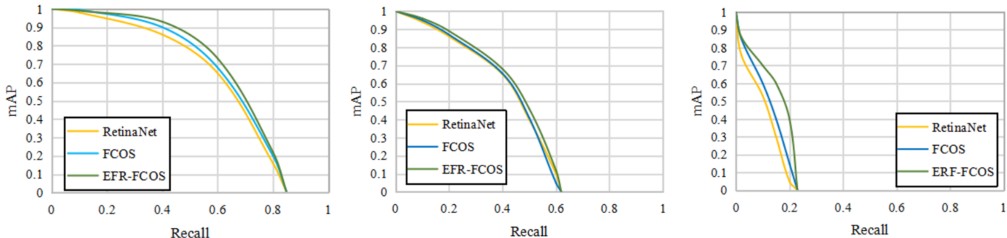

(a) precision-recall curves at IOU =0.50 (b) precision-recall curves at IOU =0.75 (c) precision-recall curves at IOU =0.90

**Figure 6** (A-C) Precision of recall curves on RetinaNet, FCOS and EFR-FCOS.

**Table 5** Alation study for different IoU thresholds on COCO 2017.

| Model | IoU | AP | $AP_{50}$ | $AP_{75}$ | $AP_S$ | $AP_M$ | $AP_L$ |
|---|---|---|---|---|---|---|---|
| FCOS-EnHead | 0.5 | 43.6 | 63.0 | 47.0 | 26.7 | 46.9 | 53.4 |
| FCOS-EnHead | 0.6 | **45.2** | **64.5** | **48.6** | **27.7** | **47.6** | **58.7** |
| FCOS-EnHead | 0.7 | 44.3 | 63.2 | 46.9 | 26.3 | 46.1 | 56.9 |

Notes.
Bold denotes the maximum of each metric in the experiment.

**Table 6** The comparisons of computational resource and runtime.

| Model | FLOPs (G) | Params (M) | AP |
|---|---|---|---|
| FCOS v2 | **234.3** | **50.9** | 43.2 |
| NAS-FCOS | 254 | 57.3 | 43 |
| TSP-FCOS | 255 | 57.5 | 44.4 |
| EFR-FCOS | 258 | 58 | **48.1** |

Notes.
Bold denotes the maximum of each metric in the experiment.

is that the sampling method has not changed. When the threshold increased to 0.6 in the second stage, we observed an increase in AP from 43.2% to 45.2%. In addition, a higher IoU threshold of 0.7 was attempted in the second stage, but AP slightly decreased. Perhaps the higher the IoU threshold, the higher the quality of the training samples, while the smaller the quantity.

Table 6 shows the comparisons of computational resource and runtime, all models adopt the backbone network with 101 layers. From that we can observe that the proposed model greatly improves the performance of the detector with slight increasing in computational resource, which indicates the effectiveness of the proposed methods for object detection.

## Qualitative examples

Figure 7 show the visualization of examples with various detectors training on COCO 2017, EFR-FCOS uses the 101 layers of GANet-GCB, the other detectors use ResNet-101. From which we can observer that the proposed EFR-FCOS (refer to 'EFR-FCOS') is superior to the popular detectors at detecting different objects such as humans and animals, etc. This is reason that EFR-FCOS uses the proposed networks, *i.e.,* GANet, AFF-FPN and EnHead to

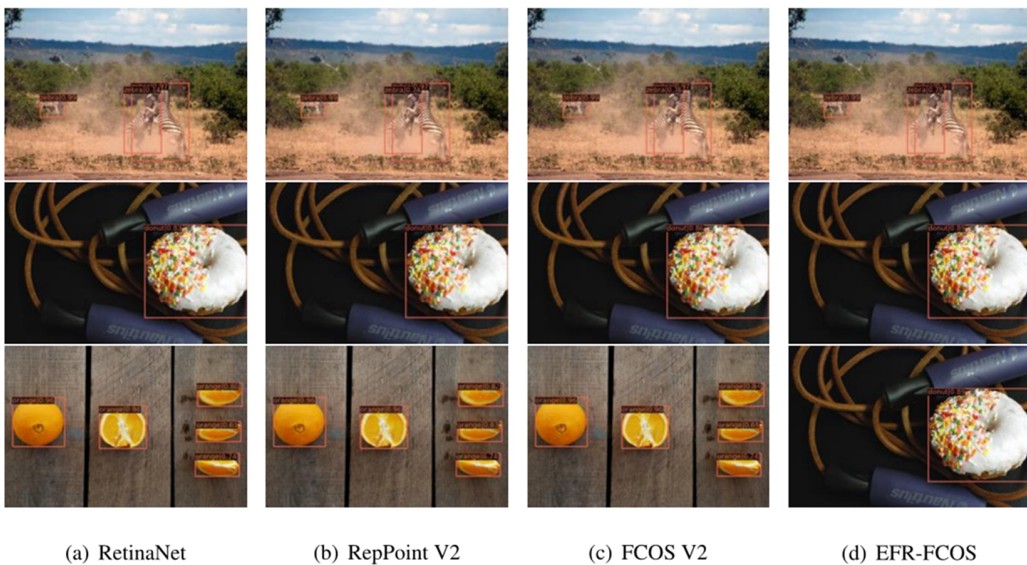

(a) RetinaNet    (b) RepPoint V2    (c) FCOS V2    (d) EFR-FCOS

**Figure 7** (A-D) Visualization of the examples from training some methods on the COCO.

improve stably the performance of detection. In particular, GANet can achieve abundant global features from images, AFF-FPN uses the dilated convolution modules with different receptive fields to fuse the features of the current layer, 3×3 convolution features, and the downsampling features of the previous layer. In addition, EnHead eliminates many false positives and regresses more accurate box boundaries to improve the accuracy of detection effectively.

## CONCLUSION

In this paper, we propose an enhancing feature reuse for fully convolutional one-stage object detection to make improvements on the three components of object detector. In the terms of backbone, we build a global attention network to collect salient information and gain global features from feature maps for performance. In the terms of neck, we build an aggregate feature fusion pyramid network to fuse multi-scale feature information by extracting global context from feature maps, which can reduce information attenuation during feature fusion. In the terms of head, we build an enhancing feature reuse head for the anchor-free one-stage detector, which uses refining the bounding boxes to improve the confidence of regression. In particular, the proposed GANet can be further researched, *i.e.,* enhancing the feature extraction capability of the backbone network by strengthening skip connections, and with adding a few number of parameters, can be applied to real-time object detection. In addition, AFF-FPN can increase feature fusion by extracting global context from feature map internally, which is also a new approach to feature fusion. The comprehensive experimental results conducted on benchmark models and datasets show that EFR-FCOS outperforms the state-of-the-art methods and achieves better performance in object detection.

### Funding

This work was supported by (i) the Guangdong Philosophy and Social Sciences Planning Project (GD21CYj21) in China and (ii) the "4th Five-Year" Plan for Educational Science in Shenzhen of China: 2023 Annual Projects (rgzn23021). There was no additional external funding received for this study. The funders had no role in study design, data collection and analysis, decision to publish, or preparation of the manuscript.

### Grant Disclosures

The following grant information was disclosed by the authors:
Guangdong Philosophy and Social Sciences PlanningProject in China: GD21CYj21.
"4th Five-Year" Plan for Educational Science in Shenzhen of China: 2023 Annual Projects: rgzn23021.

### Competing Interests

The authors declare there are no competing interests.

### Author Contributions

- Yongwei Liao conceived and designed the experiments, authored or reviewed drafts of the article, and approved the final draft.
- Zhenjun Li performed the experiments, authored or reviewed drafts of the article, and approved the final draft.
- Wenlong Feng performed the computation work, prepared figures and/or tables, and approved the final draft.
- Yibin Zhang analyzed the data, performed the computation work, authored or reviewed drafts of the article, and approved the final draft.
- Bing Zhou analyzed the data, prepared figures and/or tables, and approved the final draft.

### Data Availability

The COCO 2017 dataset is available at: https://cocodataset.org#detection-2017.
The code is available in the Supplemental File.

### Supplemental Information

Supplemental information for this article can be found online at http://dx.doi.org/10.7717/peerj-cs.2470#supplemental-information.

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
