# Peer review of "EFR-FCOS: enhancing feature reuse for anchor-free object detector"

_PeerJ Computer Science, doi:10.7717/peerj-cs.2470_

## Round 0.1 · original submission · Major Revisions

Please consider the comments carefully and revise the article accordingly. Then it will evaluated again by the reviewers.

Reviewer 1 ·

Basic reporting

1、There might be a lack of a thorough comparative analysis with state-of-the-art object detectors, particularly in terms of computational efficiency and real-world application.

2、The methodology may benefit from more detailed descriptions of the experimental setup, including data preprocessing, model training, and validation procedures.

3、The paper should ensure that the statistical analysis of results is robust, employing appropriate tests and confidence intervals to support the conclusions.

Experimental design

no comment

Validity of the findings

no comment

Additional comments

no comment

Cite this review as

Reviewer 2 ·

Basic reporting

Literature references and background:
The paper provides a good overview of related work and background context. However, some of the literature references could be more specific. For instance, when discussing improvements on ResNet, specific citations should be provided for each mentioned network (DenseNet, ResNeXt, etc.).
Suggestion: Add more specific citations when discussing related work and ensure all claims about prior research are properly attributed.

Article structure and figures:
The article follows a standard structure for a computer science paper. However, the "Implementation Details" section could be more clearly delineated from the "Experiments" section.
The figures are relevant and well-labeled, but Figure 1 (structure of EFR-FCOS) could benefit from higher resolution and more detailed labeling of components.
Suggestion: Clearly separate the implementation details from the experimental results section. Improve the resolution and labeling of Figure 1.

Raw data sharing:
The paper mentions using the COCO 2017 dataset, which is publicly available. However, it's not clear if any additional data or code used in the experiments has been made available.
Suggestion: Clearly state whether and where any additional data or code used in the experiments has been made publicly available. If possible, please provide evidence.

Self-contained results:
The paper appears to be self-contained and presents a coherent body of work. However, some of the ablation studies could benefit from more detailed analysis of why certain configurations perform better than others.
Suggestion: Provide more in-depth analysis of the ablation study results, explaining the reasons behind performance differences.

Formal results and definitions:
While this is an applied computer science paper rather than a theoretical one, some key terms and concepts could be more formally defined. For example, a more precise definition of the GANet, AFF-FPN, and EnHead components would be helpful.
Suggestion: Provide clearer, more formal definitions of key components and techniques introduced in the paper.

Experimental design

Research question and knowledge gap:
The paper does present a clear goal of improving object detection performance, but it could more explicitly state the research questions and the specific knowledge gaps being addressed.
Suggestion: Add a paragraph in the introduction that clearly outlines:
The specific research questions being addressed (e.g., "How can we improve feature extraction in the backbone of object detectors?")
The identified knowledge gaps (e.g., "Current methods struggle with X and Y aspects of object detection")
How this research aims to fill those gaps (e.g., "We propose novel approaches to address these limitations by...")


Rigorous investigation and technical standards:
The paper presents a series of experiments and comparisons that appear to be conducted rigorously. However, there are a few areas that could be improved:
The ethical considerations of the research are not explicitly mentioned.
Some of the experimental details, such as hardware specifications and runtime comparisons, are not provided.
Suggestions: Add a brief statement about ethical considerations in AI research, even if just to state that no ethical issues were identified for this study.
Provide more details about the computational resources used and include runtime comparisons with other methods.


Methods described with sufficient detail to replicate:
While the paper provides a good overview of the proposed methods, there are some areas where more detail would be beneficial for replication:
The exact architecture details of GANet, AFF-FPN, and EnHead are not fully specified.
Some hyperparameters and training details are missing.
Suggestions:Provide detailed architecture diagrams or tables specifying the exact structure of GANet, AFF-FPN, and EnHead.
Include a more comprehensive list of hyperparameters used in training, including learning rate schedules, optimizer details, etc.
Consider providing pseudocode for key algorithms or components.
Specify any data augmentation techniques used during training.
Clearly state the software versions used (e.g., specific version of PyTorch or TensorFlow).

Additionally, to further improve replicability:

Consider making the code publicly available on a platform like GitHub and referencing it in the paper.
Provide details on any pre-processing steps applied to the COCO dataset.
If any custom evaluation metrics were used beyond standard COCO metrics, provide their exact definitions.

Validity of the findings

Impact, novelty, and replication:
The paper does present novel approaches (GANet, AFF-FPN, and EnHead) and compares them to existing methods, which is good. However, it could better articulate the rationale for these specific improvements and their benefit to the literature.
Suggestions:More clearly state the limitations of existing methods that motivated each proposed component.Explain why these particular approaches were chosen and how they address gaps in current object detection literature.If any aspects of the work involve replication or validation of existing methods, explicitly state this and explain the value added.


Underlying data:
The paper mentions using the COCO 2017 dataset, which is a standard, publicly available dataset. However, there are some shortcomings in terms of data reporting:
The exact split of the dataset used for training, validation, and testing is not clearly stated.
There's no mention of where the code or model weights are made available.
Statistical analysis of the results is limited, with no mention of confidence intervals or statistical significance tests.
Suggestions:Clearly state the exact dataset split used (e.g., number of images in train/val/test).
Provide a link to a repository containing the code and trained model weights.
Include statistical analysis of the results, such as confidence intervals or significance tests when comparing to other methods.
If any preprocessing or data augmentation was applied to the COCO dataset, provide details to ensure reproducibility.


Conclusions:
The conclusions are generally well-stated and linked to the original research goals. However, there are some areas for improvement:
The conclusions could be more explicitly tied back to the original research questions and knowledge gaps identified in the introduction.
Some claims about performance improvements could be more carefully stated, acknowledging limitations or potential confounding factors.
The paper doesn't discuss potential limitations of the proposed approach or areas for future work.
Suggestions:Restructure the conclusion to clearly address each of the main research questions or goals stated in the introduction.
Be more cautious in performance claims, using phrases like "our results suggest" rather than definitive statements, especially when differences are small.
Include a discussion of limitations of the current approach and potential directions for future research.
Avoid any causal claims unless directly supported by controlled experiments.
Consider adding a brief discussion of the broader implications of this work for the field of object detection.

Additional comments

n/a

Cite this review as

Reviewer 3 ·

Basic reporting

The overall language of the paper is clear, but some sections require further polishing and simplification to ensure that international readers can easily understand. For example, the sentences in the abstract are long and complex and could be simplified, and some technical terms need to be explained to ensure that non-specialist readers can understand them.

The introduction provides an overview of current object detection algorithms but lacks a detailed description of the specific problem and knowledge gap. It is recommended to elaborate on the limitations of current algorithms and how this research addresses these gaps.

The structure of the paper conforms to PeerJ standards, but more subheadings could be added in the methods and results sections to improve readability; The literature references are comprehensive, but some may need to be updated to include the latest research developments.

Experimental design

The research question is well-defined, relevant, and meaningful. The paper explains how the research fills an identified knowledge gap.

The investigation is rigorous, but the paper needs to detail the specific steps and parameters of the experiments to enable replication by other researchers. Also, the methods section is detailed but could be further refined, especially regarding the implementation of GANet, AFF-FPN, and EnHead.

Validity of the findings

The methods section is detailed but could be further refined, especially regarding the implementation of GANet, AFF-FPN, and EnHead. The conclusions are well-stated and linked to the research question, but the paper should also discuss the limitations and future research directions.

Additional comments

This study introduces innovative improvements in the field of object detection, particularly in feature extraction and fusion. It is recommended to discuss the potential applications and practical significance of the research to enhance its impact. Please provide more experimental results and comparative analyses.

Cite this review as

---

## Round 0.2 · accepted · Accept

Thanks to the efforts of the authors. You successfully issued the concerns of the reviewers and this version satisfied the reviewers. It may be accepted. Congrats!

Reviewer 1 ·

Basic reporting

no comment

Experimental design

no comment

Validity of the findings

no comment

Additional comments

no comment

Cite this review as

Reviewer 3 ·

Basic reporting

1. The revision shows improvement in clarity and readability, with noticeable corrections in language and grammar. The authors appear to have taken steps to polish the manuscript's language based on reviewer feedback.

2. The literature review was expanded to include more recent studies, better addressing the current state of research in the field. The authors have integrated these references more effectively into their discussion.

3. The figures remain high quality, with slightly more detailed captions as suggested in the first review. This helps in making the manuscript more self-explanatory.

Experimental design

1. The originality remains strong. The authors addressed the need for further methodological details by providing more in-depth descriptions of the model components, especially GANet and AFF-FPN.

2. Improvements were made in the explanation of the methodology. The authors added more specific implementation details, which should enhance the reproducibility of the experiments.

3. The revision includes more comprehensive details and clearer descriptions, which should help other researchers replicate the study more easily.

Validity of the findings

1. The authors provided additional discussion on the limitations of their approach, addressing a key concern from the first review. The results section was also slightly expanded to include more comparative analysis.

2. The statistical analysis remains robust. The revision includes a more detailed explanation of how the results compare with other state-of-the-art methods, highlighting the novelty and impact of the findings more clearly.

3. And the conclusions have been refined to better link the findings to broader research questions, making the implications of the study more explicit.

Additional comments

The authors addressed most of the suggestions from the first review. They expanded the literature review, added more methodological details, and provided better access to data and code. The overall presentation has improved, with clearer writing and a more logically organized flow of information. The manuscript now better aligns with the expectations for a high-quality research paper.

Cite this review as